# Perception of Medical Students on the Need for End-of-Life Care: A Q-Methodology Study

**DOI:** 10.3390/ijerph19137901

**Published:** 2022-06-28

**Authors:** Jorge Barros-Garcia-Imhof, Andrés Jiménez-Alfonso, Inés Gómez-Acebo, María Fernández-Ortiz, Jéssica Alonso-Molero, Javier Llorca, Alejandro Gonzalez-Castro, Trinidad Dierssen-Sotos

**Affiliations:** 1Faculty of Medicine, University of Cantabria, 39005 Santander, Spain; jorge.barros@alumnos.unican.es (J.B.-G.-I.); ines.gomez@unican.es (I.G.-A.); javier.llorca@unican.es (J.L.); trinidad.dierssen@unican.es (T.D.-S.); 2Servicio de Medicina Intensiva, Marqués de Valdecilla University Hospital, 39008 Santander, Spain; andresfjimenez@icloud.com (A.J.-A.); jandro120475@hotmail.com (A.G.-C.); 3Marqués de Valdecilla Health Research Institute, IDIVAL, 39011 Santander, Spain; mfo5919@gmail.com; 4Consortium for Biomedical Research in Epidemiology and Public Health (CIBER de Epidemiología Y Salud Pública-CIBERESP), 28029 Madrid, Spain

**Keywords:** end-of-life care, medical students, opinion profile, Q methodology

## Abstract

End-of-life care and the limitation of therapeutic effort are among the most controversial aspects of medical practice. Many subjective factors can influence decision-making regarding these issues. The Q methodology provides a scientific basis for the systematic study of subjectivity by identifying different thought patterns. This methodology was performed to find student profiles in 143 students at Cantabria University (Spain), who will soon deal with difficult situations related to this topic. A chi-square test was used to compare proportions. We obtained three profiles: the first seeks to ensure quality of life and attaches great importance to the patient’s wishes; the second prioritizes life extension above anything else; the third incorporates the economic perspective into medical decision-making. Those who had religious beliefs were mostly included in profile 2 (48.8% vs. 7.3% in profile 1 and 43.9% in profile 3), and those who considered that their beliefs did not influence their ethical principles, were mainly included in profile 3 (48.5% vs. 24.7% in profile 1 and 26.8% in profile 2). The different profiles on end-of-life care amongst medical students are influenced by personal factors. Increasing the clinical experience of students with terminally ill patients would contribute to the development of knowledge-based opinion profiles and would avoid reliance on personal experiences.

## 1. Introduction

In recent months, the COVID-19 crisis has hit the headlines with bioethical issues regarding health care for the elderly or those in end-of-life situations [1,2]. However, this is not a new problem. The debate about how and when we should intervene has been one of the most controversial issues of health care for a long time [3,4,5,6,7]. From an economic point of view, health system’s limited resources face a great challenge since global health spending is expected to greatly increase in the next decades [8].

The limitation of therapeutic effort consists of the decision, depending on the status and future of the patient, to not apply treatments or therapeutic procedures that will provide little benefit with respect to the suffering or agony the patient is experiencing [5]. Nevertheless, there are many conditioning factors that may influence decision making, most of them both subjective and personal. In this context, training in bioethics and health economics could play a modulating role on the behaviors and attitudes of healthcare professionals regarding end-of-life care. Although a recent systematic review [9] showed that in the last decade there have been significant improvements in palliative care education in medical schools, there are important variations between individual countries [10,11]. In addition, contact with terminal patients in clinical clerkship is still rare, even in the countries with a better level of palliative medicine educational development. Several studies show that medical students feel unprepared to help patients deal with death [12] and report a lack of comfort in caring for dying patients and their relatives [13].

In Spain, training in bioethics in medical schools is scarce [14]. Furthermore, many medical schools do not routinely cover health economics in the curriculum [15] despite it being strongly recommended [16,17].

The aim of this study was to identify medical students’ views on end-of-life care and to assess whether these views changed depending on factors related to the level of studies achieved and their background (beliefs and experience with terminal patients).

## 2. Materials and Methods

Our design involved the development of a Q-methodology study [18] to identify the different profiles on end-of-life care in a sample of medical students and the analysis of the factors associated with the different profiles identified. According to the General Medical Council guidance, a patient is “approaching the end-of-life” when they are likely to die within the next 12 months. Following this definition in our study, end-of life includes different situations in which a person is at risk of dying at some point in the year ahead.

### 2.1. Q Methodology

Q methodology provides a scientific basis for the systematic study of subjectivity, identifying similarity patterns after the classification of a set of statements by respondents, a process known as “Q sort.” [18,19].

To create a Q-methodology survey, we carried out the following steps.

### 2.2. The Statement Set

The first step consisted of obtaining a set of statements of opinion, representative of different views on situations related to the provision of end-of-life treatments.

To this aim, we carried out a review of articles evaluating opinions on end-of-life care. We selected the McHugh et al., 2015 study, addressing societal perspectives about end-of-life care since their set of statements fitted our requirements [20]. We made a further selection, considering aspects such as the cultural, legislative and health policy differences with our study settings. Statements found repetitive, ambiguous, or unclear were dismissed, resulting in a final set of 27 statements out of the 49 they initially used. In addition, we added a statement exploring opinions about euthanasia.

### 2.3. Study Population and Data Collection

Our participants were selected amongst the University of Cantabria School of Medicine students. Students from all the six years of the degree were asked to participate via email and WhatsApp. The link was active from November 2019 to January 2020.

The questionnaire had three sections. In the first, students had to complete an informed consent. The second section consisted of the Q sets explained above. The third section was composed of a set of socio-demographic questions that included age, gender and year of studies. In addition, questions about religion, personal experiences and hospital contact with terminal patients, were also included. Finally, respondents were asked about the influence they considered these factors might have on their current views and thoughts towards end-of-life care.

The questionnaire was built in an online-based platform to facilitate the handling of Q sorts (which frequently involves rethinking and changing positions between previously sorted statements) [21]. To this end, we used Easy-HtmlQ [22] (an open-source license App developed in HTML5 and JavaScript). An adaptation was made to allow the inclusion of the informed consent at the beginning and the set of socio-demographic questions at the end. The Easy-HtmlQ online platform was deployed using Netlify web hosting services [23] and Google’s Firebase tool [24] for storing data.

The Q-sort process started, first by classifying the set of statements, based on whether the student agreed, disagreed, or had a neutral opinion about them. Until a sentence was classified the respondent could not read the next.

In the next step, respondents were asked to sort and position these statements on a response grid, known as a “Q grid” [25]. They started by ranking the statements with which they had previously agreed. First, they chose those two statements with which they agreed the most (+3), followed by the four which they considered to be +2 and ending with the five statements for +1 position. The same process was done for the left side, placing the previously classified as disagree statements in the −3, −2 and −1 columns.

The remaining statements, sorted as neutral, were placed in the 0 position. However, in most cases there was not the same number of 0 spots as statements marked as neutral. If there were more than 6 neutral statements or there were more agree or disagree statements than positions for them, they had to classify the statements in the remaining places. This last step of the Q sort ensured that respondents had to double-check their previous choices.

### 2.4. Statistical Analysis

The Q-method analysis was performed in Ken-Q Analysis v1.0.3, a specific Webplication for Q Methodology [26]. First, a principal component analysis, in which the respondents are correlated in order to identify a number of natural groupings of Q sort; from here on, we refer to this natural group as “profiles”. Secondly, a Q-method analysis was developed, consisting of: (1) flagging the Q sort that defines each profile, (2) calculating the score of statements for each profile and (3) finding the distinguishing statements (those ranking in a position that significantly differed from their rank in other profiles) and consensus statements (those that are not distinguishing any profile from each other). The standard analytical process in Q methodology has been previously described in detail [27,28].

Finally, after defining the different profiles, we performed a chi-square test to identify the factors related to each profile. This statistical analysis was developed with the statistical package STATA/SE 16 (Stata Corp, College Station, TX, USA).

## 3. Results

### 3.1. Study Population

The link to the survey was distributed amongst the medical students (773 in total), obtaining 143 responses (response rate of 18.5%). The characteristics of the respondents are presented in Table 1. The mean age was 22.1 (SD: 2.68) years. About two thirds (70.6%) were in fourth, fifth or sixth year of their medical studies. Ninety-eight of the 143 (68.5%) respondents were female.

### 3.2. Selected Profiles

We selected the three profiles explaining more variance (56%); all three profiles were interpretable, so from here on we will refer them as profiles 1, 2 and 3. One hundred and twenty-one respondents (84.6%) could be included in one of these profiles and 22 (15.4%) could not be classified and were excluded from the statistical analysis.

Profiles were characterized by their distinguishing statements; from here on, each statement is presented followed by its number, its position on the Q grid (from −3 to +3), and an * in the case of being a distinguishing factor. For instance, when describing profile 1, we used (#5, 0*), which indicates statement #5 in Table 2 (“I would place more value on end-of-life treatments than many medical treatments for non-terminal conditions”) scored 0 in people with profile 1 and allowed us to distinguish this profile from profiles 2 and 3.

### 3.3. Profile 1: Students Prioritizing Patient’s Wishes and Quality of Life

The first of the three profiles accounts for 17.5% of the sample (25 respondents). Figure 1 shows the composite Q sort for this group.

People holding this view do not express preference in statements that are closely related to wishes or perceptions of the terminal patients or their relatives: “Extending life for people with terminal illnesses is only postponing death” (#21, 0*) or “I would place more value on end-of-life treatments than many medical treatments for non-terminal conditions” (#5, 0*). They also worry about the patient’s wishes, respecting their power to decide about their own life: “If somebody wants to keep fighting until the last possible moment, they should be allowed to do so, regardless of cost” (#3, 2).

They do not defend life at any cost: they score negatively in “Life is sacred and if it is possible to preserve life, every effort should be made to do so” (#22, −3) and “It is human nature to want to preserve life and extend it as long as we can—it is one of our most basic instincts” (#7, −1*). They do not encourage a terminal patient to prolong his life unless ensuring a good quality of life: “Real help and compassion should be about providing a death with dignity instead of more drugs to get a few more weeks or months out of a very sick body” (#11, 2).

This view ignores the burden of health care expenses, justifying the most effective treatment regardless of the cost. Therefore, they score those sentences related with prioritizing costs over health care as negative: “Everyone has a right to basic healthcare but there have to be limits and expensive end-of-life drugs are not basic care” (#24, −3*); “Expensive drugs for people who are terminally ill and will not benefit very much are not a good use of public funding” (#6, −2); “Treatments that are very costly in relation to their health benefits should be withheld” (#17, −2). Finally, respondents of this profile showed a special sensitivity towards terminal patients, so they marked the statement “I think life-extending treatments for people who are terminally ill are of less value as people get older” as negative (#26, −2).

### 3.4. Profile 2: Students Believing That Life Must Be Extended Whatever the Cost

Forty-two respondents (29.4%) fit in the second profile. Figure 2 shows its composite Q sort. People with this view advocate life extension always; therefore, they agree with “It is human nature to want to preserve life and go on living for as long as we can—it is one of our most basic instincts” (#7, 2*) and “If the means of helping someone live longer exists, it is morally wrong to deny them the treatment” (#15, 2*), while disagreeing with “Extending life for people with terminal illnesses is only postponing death” (#21, −3*).

They defend life at any cost: “If somebody wants to keep fighting until the last possible moment, they should be allowed to do so, regardless of cost” (#3, 2). In contrast to the other views, they reject euthanasia since it is presented as a measure of health expenditure control, and this is not a primary concern for them: “An objective measure of health expenditure control could be to legalize the euthanasia process” (#28, −1*).

Consequently, respondents included in this profile maintain that every effort should be made to prolong life, even if it is for a short time: “It may not sound like much, but a few extra weeks or months might mean an awful lot to a family affected by a terminal illness” (#9, 3*); “It is important to provide life-extending treatments to give a dying person time to reach a significant milestone, such as a family event or a personal achievement” (#25, 1*); and, thus, they despise statements undervaluing life vs. costs: “Treatments that provide short life extension are not worth it—they are only prolonging the pain for the patient’s family/friends” (#20, −3*); “End-of-life drugs are not a cure, they are life-prolonging. There is no point in delaying the inevitable for a short time” (#18, −2*); “Patients at the end of life will grasp any slightest hope but that is not a good reason for the NHS to provide costly treatments that may extend life by a short time” (#19, −2*).

Similar to profile 1, they consider terminal patients’ worth special attention, though they accept that important health gains are not expected despite greater spending; therefore, they disagree with: “Treatments that are very costly in relation to their health benefits should be withheld” (#17, −2); “I think life-extending treatments for people who are terminally ill are of less value as people get older” (#26, −2).

The fact that students with this profile do not reject sentences such as “Not giving access to life-extending medicine to a person with a terminal illness is the same as killing them” (#16, 0*); “Life is sacred and if it is possible to preserve life, every effort should be made to do so” (#22, 0*) is another indicator of how much they respect life. These two sentences distinguish this profile as students in profiles 1 and 3 score them negatively.

Respondents included in this profile do not show a clear position in those statements evaluating expected quality of life of a terminal patient. They do not agree with sentences such as “At the end of their life, patients should be cared for at home, with a better quality of life, rather than have aggressive and expensive treatments that will only extend life for a short period of time” (#2, 0*); “Life should only be extended if the patient’s quality of life during that time will be good” (#10, 0*); “Real help and compassion should be about providing a death with dignity instead of more drugs to get a few more weeks or months out of a very sick body” (#11, 0*), suggesting that they prioritize life extension over quality of life. These three sentences distinguish profile 2 as students in profiles 1 and 3 score them positively.

Finally, this profile does not show a clear position when the moral problem of prolonging the life of a terminally ill patient is assessed (“To extend life in a way that is beneficial to the patient is morally the right thing to do”) (#14, 0*). In contrast, respondents in the other two profiles strongly disagree with this statement.

### 3.5. Profile 3: Students Maximizing Health Benefits and Economic Aspects

The third profile includes most respondents, accounting for 37.8% of the sample (54 participants). Figure 3 shows the composite Q sort for this profile. 

The main difference from the other two profiles is that they consider that the Public Health System should prioritize the cost of medical care: “Patients at the end of life will grasp any slightest hope but that is not a good reason for the NHS to provide costly treatments that may extend life by a short time” (#19, 1*). Therefore, they negatively score sentences such as “I would place more value on end-of-life treatments than on many medical treatments for non-terminal conditions” (#5, −2*) and “If somebody wants to keep fighting until the last possible moment, they should be allowed to do so, regardless of cost” (#3, −1*). They would support a patient’s will to end life: “Patients should have the right to refuse life-extending treatments if they choose” (#1, 3*). In addition, they do not express a preference regarding sentences that imply a greater expenditure in end-of-life care: “If a life-extending treatment for terminally ill patients is expensive, but is the only treatment available, it should still be provided” (#8, 0*); “We should spend proportionately more on patients when we feel those patients have not had their fair innings—in terms of the length of their life or the quality of that life” (#13, 0*); “Treating people at the end of life is not going to result in big health gains but the health system should be about looking after those patients in greatest need” (#27, 0*).

Unlike respondents with profile 2, people included in this profile strongly reject the extension of life just to keep the patient alive, disagreeing with the statements which advocate this idea: “Life is sacred and if it is possible to preserve life, every effort should be made to do so” (#22, −3); “A year of life is of equal value for everyone” (#12, −3*); “Not giving access to life-extending medicine to a person with a terminal illness is the same as killing them” (#16, −2*); “To extend life in a way that is beneficial to the patient is morally the right thing to do” (#14, −2). On the other hand, respondents in this group present some similarities with group 1, since they consider that life extension is only worthwhile and ethical if it results in actual health gains, not just stopping death from happening: “I would not want my life to be extended just for the sake of it—just keeping breathing is not life” (#23, 3); and provided quality of life will be good: “Life should only be extended if the patient’s quality of life during that time will be good” (#10, 2); “Real help and compassion should be about providing a death with dignity instead of more drugs to get a few more weeks or months out of a very sick body” (#11, 2).

### 3.6. Characteristics Associated with the Profiles

Respondents’ characteristics associated with the profiles are presented in Table 3.

The only two variables significantly associated with the profiles were the respondents’ religious beliefs and the influence they considered these beliefs have on their ethical principles. Those students who profess religious beliefs are included in profile 2 in greater proportion (48.8% vs. 7.3% in profile 1 and 43.9% in profile 3), while those who consider their beliefs do not influence their ethical principles are included in a greater proportion in profile 3 (48.5% vs. 24.7% in profile 1 and 26.8% in profile 2).

It should be noted that profile 3 was the least numerous in students who believed that their previous personal experience with a close relationship with terminally ill patients conditioned their position (18.2%). On the contrary, this third profile was predominant in those who did not have this experience or believed that, despite having the experience, they were never influenced by it (52.2%). No differences associated to sex, age, year of studies, average grade, previous contact with terminally ill patients or living with their family were found.

## 4. Discussion

Our study identified three main profiles amongst medical students towards end-of-life care. The first view (students prioritizing patient’s wishes and quality of life) leaves health economics in the background and focuses on respecting the patient’s choice regarding his/her own life, provided quality of life is good. The second profile (students believing that life must be extended whatever the cost) is similar to the previous one regarding economic aspects—no sparing of expenses on treatments for terminally ill patients. However, people with this view think every patient should have the chance to prolong his/her life to the limit, leaving aside quality of life, and they do not support the limitation of therapeutic effort. The last profile (students maximizing health benefits and economic aspects) differs from the previous two in that it incorporates the economic perspective into medical decision-making, prioritizing healthcare investment in those patients with the possibility of recovery.

Other studies assessing end-of-life care beliefs [20,29,30] identified different profiles. Similar to our study, McHugh et al. [20], in a study in the UK with 59 Q-set respondents, described a first profile in which “patients’ rights are central and life is regarded as precious and priceless. So, even high cost treatments that deliver limited benefits should not be withheld from patients”, and a second group whose main view was “to achieve the greatest health gains for the greatest number through the efficient allocation of limited resources”, which clearly corresponds to our profiles 2 and 3. They also described a third profile for whom quality of life was one of their main concerns, similar to our first profile, but, in our study, this profile prioritizes the patient’s willingness—justifying the provision of any possible treatment if asked for. Furthermore, the third profile identified by McHugh et al. values more economic aspects not taken into account in our first profile [20]. In a study performed in the Netherlands, Wouters et al. [29] also described three profiles from 46 Q sorts of the general population. Two were similar to our profiles 1 (“the care that terminal patients receive should at all times respect the patients’ quality of life and dignity”) and 3 (“priority should be given to treatments that generate the most health and patients who benefit most from treatment”), which focus on cost-effectiveness as the main criterion for decision-making. Their remaining profile emphasized “the importance of equality in opportunities and hence access to healthcare” and “denies giving priority in any circumstance, because assigning priority to some patients at the cost of other conflicts with every person’s basic and equal right to healthcare”. Finally, Van Exel et al. [30], in their multinational study identified five viewpoints from 329 members of the general public, two of them being very similar to our profiles 1(“quality of life is more important than simply staying alive”) and 2 (“the intrinsic value of life and healthy living”). However, our third profile could be interpreted as a mixture of two of the views identified by Van Exel: “severity and the magnitude of health gains” and “fair innings, young people and maximizing health benefits”, based on the magnitude of health gains and the relevance of a patient’s age in priority setting. The last profile identified in the Van Exel study (“egalitarianism, entitlement and equality of access”) takes into account economic aspects in the access to medical attention that, given the characteristic of the Spanish National Health System (based on the principles of universal access), have not been included in our study. The fact that their sample was composed of people from ten different countries and the wide range of their scores may be the reason why they described five different profiles.

In our study, religious issues were the only aspect significantly associated with a profile. Respondents who claimed to be religious were found to be more likely included in the second profile (students believing that life must be extended whatever the cost) than in the other two. In this regard, only those amongst this group who were practicing and considered their religious beliefs to influence their view on end-of-life care were more likely to classify in the second profile, whereas those who considered themselves as not influenced by religion were predominantly included in the third profile (students maximizing health benefits and economic aspects). Only one study using Q methodology analyzed the influence of religion and suggested that participants who place a high value on life might be influenced by their religious beliefs [30]. Along the same lines, studies using other methodologies have shown that medical students who had active religious beliefs were likely to disagree with actions that hasten death [31].

Our study also highlights the importance of having a personal experience with terminally ill patients on the perception of medical students. Those respondents who did not have a close relationship with the terminally ill or believed themselves not to be influenced by such an experience were predominantly classified in the third profile, similar to a study assessing knowledge and attitudes about end-of-life care in community health care providers that showed more positive attitudes in those who had experiences of death of relatives or friends [32]. Furthermore, it has been suggested that being a final-year student is associated with a higher likelihood of agreeing with actions that precipitate death in terminally ill patients [31] which concurs with our third profile. The fact that more than half of the final-year students (56.5%) were included in the third profile could partially be explained by the influence of the knowledge acquired throughout their degree. It has been suggested that a formal curriculum in health systems and health policy should be an essential component of medical education, and that students should be provided with this training prior to their third year, so that they can integrate their clinical experiences into a broader framework [33].

Lastly, another aspect to take into account is the influence of clinical experience. Students about to complete their medical degree spend most of the year performing their clinical clerkship. This first contact with the National Health System has been suggested to be the most powerful factor influencing self-perceived attitudes towards end-of-life care [34]. We found that previous contact with terminal patients in clinical clerkships favored being included in profile 1 in a greater proportion than those who did not have this experience. However, the fact that only about 38% of the respondents in the last year of the degree had contact with terminally ill patients during their clerkship, could be a possible explanation of the low prevalence of the first profile in this subgroup of students. In this respect, a more in-depth, standardized practical training for medical and nursing students is considered to be necessary in order to ensure an adequate end-of-life care provision, especially regarding the humane component and empathy that professionals must show [35]. In addition, a previous study assessing medical students’ views on the development of empathic behavior revealed that early patient contact and clinical skills and communication courses were found by the students to be very helpful to foster their ability to empathize with patients [36].

There are some limitations in our study. First, the low participation rates of students in the first years prevented us from exploring how being in a particular year of the degree determined the profile. However, we were able to separately analyze the role of clinical experience, because more than half the students in the final year of the degree participated in the survey. Secondly, statistical power was limited for the analyses of associated factors, due to the small number of exposures in some of the different subgroups. However, this issue did not affect the Q-methodology analysis. Despite the low response rate, the characteristics of the study population were not altered in the sample, since the proportion of women was similar (68.5% in the sample vs. 71.7% in the population) and we obtained questionnaires from respondents of 27 different Spanish provinces.

To the best of our knowledge, this is the first study evaluating opinions on bioethics and end-of-life care applying Q methodology to a sample composed exclusively of medical students. This allows an in-depth analysis of the profiles of medical students on a matter of crucial importance on their future professional life. Previous studies used more heterogeneous samples, assessing students’ views but also those of others members of the public, such as researchers and health policy makers, which may mask the students’ opinion on the matter [37]. Different studies have highlighted the need for further research on students’ perception of end-of-life care, as well as the medical schools’ approach to the subject [29,38,39,40]. Although other methodologies have proven to be useful for studying public views on end-of-life care and health economics [41], such as budget allocation [42,43] or willingness-to-pay [44,45], we picked Q methodology since we wanted to obtain a broader view of the different patterns of opinion amongst the students. Achieving this goal would have been difficult using any of the other methodologies, considering that they mostly explore specific scenarios.

## 5. Conclusions

We found three different profiles on end-of-life care amongst medical students. These profiles are influenced by personal factors (beliefs, experience with terminally ill patients). We propose that increasing the clinical experience of students with terminally ill patients and reinforcing bioethics and health economics, would contribute to the development of more knowledgeable opinion profiles, and avoid relying on personal experiences.

## Figures and Tables

**Figure 1 ijerph-19-07901-f001:**
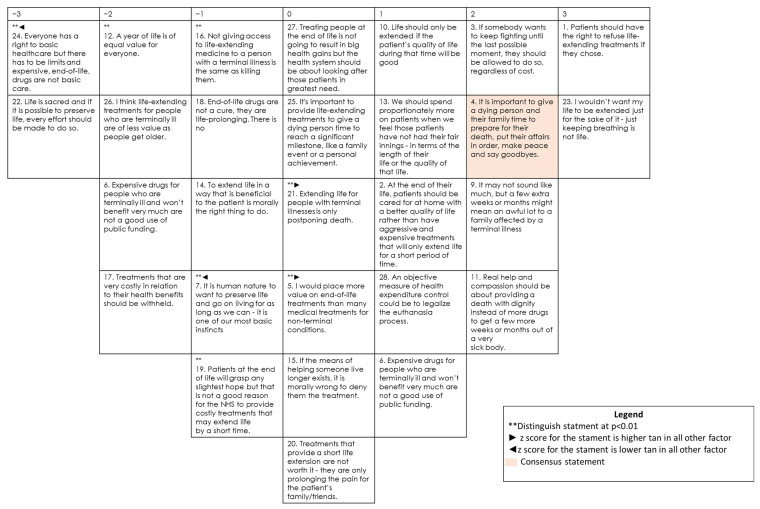
Composite Q sort for factor 1.

**Figure 2 ijerph-19-07901-f002:**
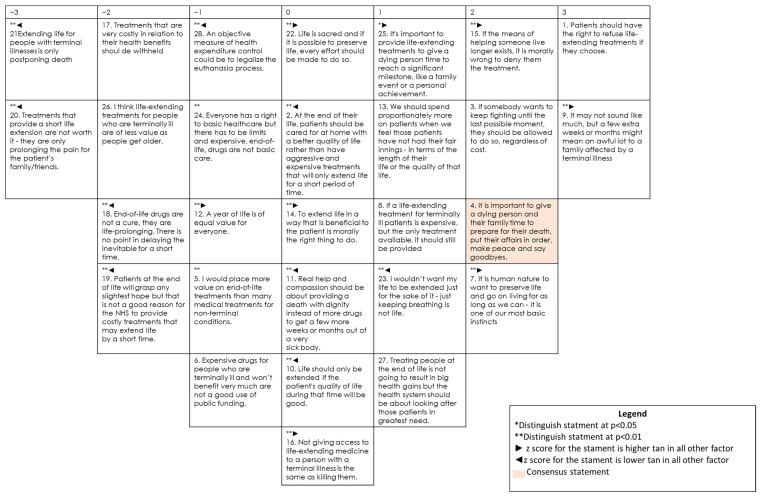
Composite Q sort for factor 2.

**Figure 3 ijerph-19-07901-f003:**
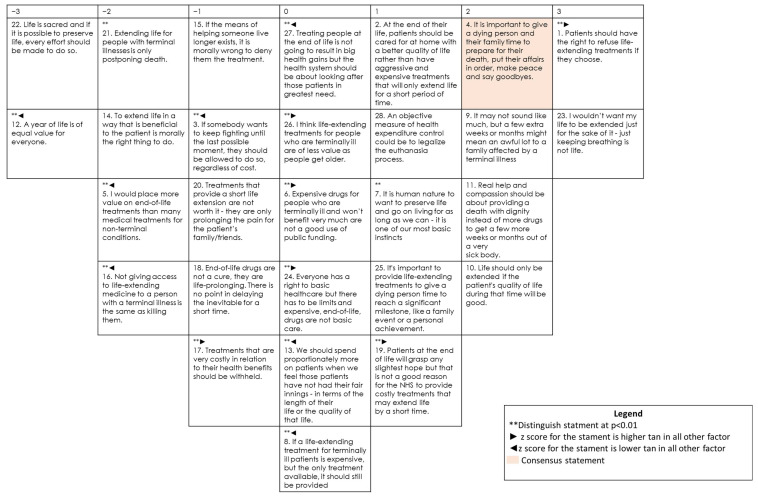
Composite Q sort for factor 3.

**Table 1 ijerph-19-07901-t001:** Sample characteristics.

Variable	Category	All Respondents (143)	Respondents with Profile (121)	Respondents without Profile (22)
		N (%)	N (%)	N(%)
**Gender**	Male	45 (31.47)	39 (32.23)	6 (27.27)
	Female	98 (68.53)	82 (67.77)	16 (72.73)
**Year of studies**	First to third year students	42 (29.37)	37 (30.58)	5 (22.73)
	Fourth to fifth-year students	48 (33.57)	38 (31.40)	10 (45.45)
	Sixth-year students	53 (37.06)	46 (38.02)	7 (31.82)
**Starting year**	2013	8 (5.59)	6 (4.96)	2 (9.09)
	2014	50 (34.97)	44 (36.36)	6 (27.27)
	2015	26 (18.18)	22 (18.18)	4 (18.18)
	2016	15 (10.49)	11 (9.09)	4 (18.18)
	2017	19 (13.29)	17 (14.05)	2 (9.09)
	2018	15 (10.49)	11 (9.09)	4 (18.18)
	2019	10 (6.99)	10 (8.26)	0 (0.00)
**Autonomous community**	Cantabria	78 (54.55)	64 (52.89)	14 (63.64)
	Asturias	20 (13.99)	18 (14.88)	2 (9.09)
	Castile and Leon	13 (9.09)	10 (8.26)	3 (13.64)
	Andalusia	7 (4.90)	7 (5.79)	0 (0.00)
	Madrid	7 (4.90)	6 (4.96)	1 (4.55)
	Basque Country	4 (2.80)	4 (3.31)	0 (0.00)
	Castile La Mancha	4 (2.80)	4 (3.31)	0 (0.00)
	Others	10 (6.99)	8 (6.61)	2 (9.09)
**Age, (mean (SD))**		22.13 (2.68)	21.97 (2.37)	23.05 (3.96)

**Table 2 ijerph-19-07901-t002:** Factor scores per statements.

Statement	F1	F2	F3
1. Patients should have the right to refuse life-extending treatments if they choose.	3	3	**3***
2. At the end of their life, patients should be cared for at home with a better quality of life rather than have aggressive and expensive treatments that will only extend life for a short period of time.	1	**0***	1
3. If somebody wants to keep fighting until the last possible moment, they should be allowed to do so, regardless of cost.	2	2	**−1***
4. It is important to give a dying person and their family time to prepare for their death, put their affairs in order, make peace and say goodbyes.	2	2	2
5. I would place more value on end-of-life treatments than many medical treatments for non-terminal conditions.	**0***	−1	**−2***
6. Expensive drugs for people who are terminally ill and will not benefit very much are not a good use of public funding.	−2	−1	**0***
7. It is human nature to want to preserve life and go on living for as long as we can—it is one of our most basic instincts.	**−1***	**2***	1
8. If a life-extending treatment for terminally ill patients is expensive, but the only treatment available, it should still be provided.	1	1	**0***
9. It may not sound like much, but a few extra weeks or months might mean an awful lot to a family affected by a terminal illness.	2	**3***	2
10. Life should only be extended if the patient’s quality of life during that time will be good.	1	**0***	2
11. Real help and compassion should be about providing a death with dignity instead of more drugs to get a few more weeks or months out of a very sick body.	2	**0***	2
12. A year of life is of equal value for everyone.	−2	−1	**−3***
13. We should spend proportionately more on patients when we feel those patients have not had their fair innings—in terms of the length of their life or the quality of that life.	1	1	**0***
14. To extend life in a way that is beneficial to the patient is morally the right thing to do.	−1	**0***	−2
15. If the means of helping someone live longer exists, it is morally wrong to deny them the treatment.	0	**2***	−1
16. Not giving access to life-extending medicine to a person with a terminal illness is the same as killing them.	−1	**0***	**−2***
17. Treatments that are very costly in relation to their health benefits should be withheld.	−2	−2	−1
18. End-of-life drugs are not a cure, they are life-prolonging. There is no point in delaying the inevitable for a short time.	−1	**−2***	−1
19. Patients at the end of life will grasp any slightest hope but that is not a good reason for the NHS to provide costly treatments that may extend life by a short time.	−1	**−2***	**1***
20. Treatments that provide a short life extension are not worth it—they are only prolonging the pain for the patient’s family/friends.	0	**−3***	−1
21. Extending life for people with terminal illnesses is only postponing death.	**0***	**−3***	−2
22. Life is sacred and if it is possible to preserve life, every effort should be made to do so.	−3	**0***	−3
23. I would not want my life to be extended just for the sake of it—just keeping breathing is not life.	3	1	3
24. Everyone has a right to basic healthcare but there have to be limits and expensive, end-of-life, drugs are not basic care.	**−3***	−1	**0***
25. It is important to provide life-extending treatments to give a dying person time to reach a significant milestone, such as a family event or a personal achievement.	0	**1***	1
26. I think life-extending treatments for people who are terminally ill are of less value as people get older.	−2	−2	**0***
27. Treating people at the end of life is not going to result in big health gains but the health system should be about looking after those patients in greatest need.	0	1	**0***
28. An objective measure of health expenditure control could be to legalize the euthanasia process.	1	**−1***	1

Statement #4: “It is important to give a dying person and their family time to prepare for their death, put their affairs in order, make peace and say goodbyes” was identified as a consensus statement. The three profiles agreed in this statement with +2.

**Table 3 ijerph-19-07901-t003:** Respondents’ characteristics associated with the profile.

		Profile				
Variable	Category	*Profile 1: Students Prioritizing Patient’s Will and Quality of Life*	*Profile 2: Students Believing That Life Must Be Extended Whatever the Cost*	*Profile 3: Students Maximizing Health Benefits and Economic Aspects*	Chi-Square	*p*-Value
Year of studies	First to third year students n (%)	9 (24.32)	15 (40.54)	13 (35.14)		0.258
	Fourth to fifth year students n (%)	7 (18.42)	16 (42.11)	15 (39.47)	5.297	
	Sixth year students n (%)	9 (19.57)	11 (23.91)	26 (56.52)		
Average grade *	5 < 7 n (%)	5 (15.63)	13 (40.63)	14 (43.75)		0.444
	7–8 n (%)	14 (25.00)	19 (33.93)	23 (41.07)	3.730	
	>8 n (%)	4 (16.00)	6 (24.00)	15 (60.00)		
Previous contact with terminally ill patients in clinical clerkship	none or scarce n (%)	14 (17.50)	28 (35.00)	38 (47.50)		0.634
	some or frequent n (%)	7 (25.00)	10 (35.71)	11 (39.29)	0.912	
Lives with family	No n (%)	12 (19.35)	25 (40.32)	25 (40.32)		0.409
	Yes n (%)	13 (22.03)	17 (28.81)	29 (49.15)	1.787	
Both parents with university studies	No n (%)	12 (18.46)	20 (30.77)	33 (50.77)		0.342
	Yes n (%)	13 (23.21)	22 (39.29)	21 (37.50)	2.1443	
Religious beliefs	No n (%)	22 (27.50)	22 (27.50)	36 (45.00)		0.012
	Yes n (%)	3 (7.32)	20 (48.78)	18 (43.90)	8.888	
Influence of religious beliefs in ethical principles	Never/scarce/no/not applicable/n (%)	24 (24.74)	26 (26.80)	47 (48.45)		0.001
	Yes/sometimes/always n (%)	1 (4.17)	16 (66.67)	7 (29.17)	14.353	
Influence of personal experience with terminal patient in EoL care opinion	Never/not applicable n (%)	12 (17.91)	20 (29.85)	35 (52.24)		0.064
	Sometimes n (%)	5 (15.63)	12 (37.50)	15 (46.88)	8.876	
	Always n (%)	8 (36.36)	10 (45.45)	4 (18.18)		
Gender	Male n (%)	8 (20.51)	13 (33.33)	18 (46.15)		0.969
	Female n (%)	17 (20.73)	29 (35.37)	36 (43.90)	0.062	
Age (mean (sd))		22.42 (3.74)	21.77 (2.15)	21.91 (1.67)		0.565

* Grading in Spanish universities are 0–10. Five points are required to pass.

## Data Availability

The data that support the findings of this study are available on request from the corresponding author.

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
