# Peer review of "Perception of Medical Students on the Need for End-of-Life Care: A Q-Methodology Study"

_ijerph, 2022, doi:10.3390/ijerph19137901_

Round 1

Reviewer 1 Report

Attitude and Knowledge of QOL is important, so this paper is meaningful.

I give some comments below

1. Figure 1 is not needed. You can describe just text. 

2. For qui-square, you have to show not only p-value but also the value of qui-square in Table 3. 

Author Response

  1. Figure 1 has been eliminated
  2. Values of qui-square has been added in table 3

Reviewer 2 Report

Dear Authors,

The topic “Perception of medical students on the need for end-of-life care. A Q-methodology Study” is  interesting,

There are some suggestions for authors,

1.         Du to the topic is Perception of medical students on the need for end-of-life, please add identifying related end-of-life or address more why focus elder patients do not focus on cancer patients or palliative care?

2.         Page 2, “Our participants were selected amongst the University of Cantabria School of Medicine students. Students from all the six years of the degree were asked to participate via email and WhatsApp” the study’s participants how to mentation Students all get related end-of-life care based line knowledge, please clarify.

3.         age 2, line 45, “The aim of this study was to identify medical students’ views on end-of-life care and to assess whether these views changed depending on factors related to the level of studies achieved and their background” for more contributes, suggest authors address more the study’s aim via specific end-of-life care patients? Is focused on end-of-life care with adult, palliative care, or elderly patients’ issues?

4.         Page 2, line 89, please add references support, “known as “Q-grid” (Figure 1).”

5.         Study population, why response rate is only 18.5%?

6.         Page 7, figure 2 is Vague.

Thank you.

Author Response

Thank you so much for your comments. Below I indicate the changes made based on your suggestions:

  1. Due to the topic is Perception of medical students on the need for end-of-life, please add identifying related end-of-life or address more why focus elder patients do not focus on cancer patients or palliative care?

Following the reviewer's suggestion we have added a paragraph in order to clarify that end-of-life patients included in our study, a heterogeneous group of patients with different illnesses, not only elderly people:

Page 2 line 58: According to the General Medical Council guidance a patient is ‘approaching the end-of-life’ when they are likely to die within the next 12 months. Following this definition in our study End-of life includes different situations in which a person is at risk of dying at some point in the year ahead.

  1. Page 2, “Our participants were selected amongst the University of Cantabria School of Medicine students. Students from all the six years of the degree were asked to participate via email and WhatsApp” the study´s participants how to mentation?? Students all get related end-of-life care based line knowledge, please clarify.

Our study includes medical students with different levels of knowledge and experience with terminal patients.  In order to clarify this aspect, the following paragraph has been included in the results section:

Thirty-two (25%) of the 143 students surveyed confirmed having had previous contact with terminally ill patients during their clinical clerkship, and 81 (56%) had a personal or close relationship (a friend, a relative...) with terminally ill patients.

  1. The aim of this study was to identify medical students’ views on end-of-life care and to assess whether these views changed depending on factors related to the level of studies achieved and their background” for more contributes, suggest author address more the study´s aim via specific end-of.life care patients?

Unfortunately, we have not separately analysed different causes of end-of-life. However, it is a very interesting suggestion that could be studied in future research.

4.Page 2, line 89 please add references support, “know as “Q´grid” (Figure 1)”

 Following recommendation of the  Reviewer we have added a reference support: Watts, S.; Stenner, P. Doing Q Methodological Research: Theory, Method and Interpretation; SAGE Publications Ltd: 1 Oliver’s Yard, 55 City Road, London EC1Y 1SP United Kingdom, 2012; ISBN 9781849204156.

  1. Study population, why response rate is only 18.5%?

Our response rate could be explained out of several reasons; firstly, both the time needed to complete the survey (more than 20 minutes) and the high level of attention required to classify the statements and classify them in the Q grid may have been barriers to complete the survey successfully. Secondly, the fact that incentives were not used to promote participation may also have played a role. Finally, several studies show that online surveys achieve lower response rates than other methods (such as telephone or postal survey) (VanGeest, J. B., Johnson, T. P., & Welch, V. L. (2007). Methodologies for improving response rates in surveys of physicians: A systematic review. Evaluation &the Health Professions, 30, 303-321).

  1. Figure 2 is vague

This figure has been replaced with another of higher quality.

Reviewer 3 Report

Extremely interesting study with implications both for practice and for education itself in terms of the training of these professionals.

Tracing these profiles in such a systematic and profound way allows for a representative picture of end-of-life care.

I would say that the authors could add a little more evidence in the introduction to support the relevance and pertinence of the study. The discussion, in turn, is rich in information and allows a global view of the results obtained in the light of the currently existing evidence.

The results seem to me to be quite thorough.

In the conclusion, the implications for practice, research, and education could be explicit.

Good job.

Author Response

Thank you so much for your comments. Following your recommendations, we have added a paragraph in the introduction section to support the relevance of the study:

The limitation of therapeutic effort consists in deciding, depending on the status and future of the patient, to not apply treatments or therapeutic procedures that will provide little benefit with respect to the suffering or agony the patient is experiencing.[5] Nevertheless, there are many conditioning factors that may influence decision making, most of them both subjective and personal. In this context, training in bioethics and health economics could play a modulating role on the behaviours and attitudes of healthcare professionals regarding end-of-life care. Although a recent systematic review [1] shows that in the last decade there have been significant improvements in palliative care education in medical schools,  there are important variations between individual countries [2,3]. In addition, contact with terminal patients in clinical clerkship is still scarce even in the countries with a better level of Palliative Medicine educational development. Several studies show that medical students feel unprepared to help patients deal with death [4] and report a lack of comfort in caring for dying patients and their relatives [5].